# Delay and Energy Consumption of MQTT over QUIC: An Empirical Characterization Using Commercial-Off-The-Shelf Devices

**DOI:** 10.3390/s22103694

**Published:** 2022-05-12

**Authors:** Sidna Jeddou, Fátima Fernández, Luis Diez, Amine Baina, Najid Abdallah, Ramón Agüero

**Affiliations:** 1Department of Communication Systems, National Institute of Posts and Telecommunications, Rabat 10112, Morocco; baina@inpt.ac.ma (A.B.); najid@inpt.ac.ma (N.A.); 2Ikerlan Technology Research Centre, Basque Research Technology Alliance (BRTA), 20500 Arrasate/Mondragón, Spain; ffernandez@ikerlan.es; 3Department of Communication Engineering, Universidad de Cantabria, 39005 Santander, Spain; ldiez@tlmat.unican.es (L.D.); ramon@tlmat.unican.es (R.A.)

**Keywords:** Internet of Things (IoT), Quic UDP Internet Connections (QUIC), Transmission Control Protocol (TCP), Message Queuing Telemetry Transport (MQTT), Quality of Service (QoS), energy consumption

## Abstract

The QUIC protocol, which was originally proposed by Google, has recently gained a remarkable presence. Although it has been shown to outperform TCP over a wide range of scenarios, there exist some doubts on whether it might be an appropriate transport protocol for IoT. In this paper, we specifically tackle this question, by means of an evaluation carried out over a real platform. In particular, we conduct a thorough characterization of the performance of the MQTT protocol, when used over TCP and QUIC. We deploy a real testbed, using commercial off-the-shelf devices, and we analyze two of the most important key performance indicators for IoT: delay and energy consumption. The results evince that QUIC does not only yield a notable decrease in the delay and its variability, over various wireless technologies and channel conditions, but it does not hinder the energy consumption.

## 1. Introduction

In the last few years, we have witnessed a strong increase in Internet of Things (IoT)-based services, having varying characteristics and so posing different requirements to communication protocols. In particular, low latency and energy efficiency are some of the most common challenges of IoT services. Within the IoT field, Industrial IoT (IIoT) is becoming one of the most frequently discussed topics. In fact, it has a notable impact on the economy, since it has the potential to increase industrial productivity by 30% [1]. In 2011, the German government introduced the term Industry 4.0, referring to the fourth industrial revolution [2]. This idea is based on four concepts: interconnection, information transparency, decentralized decisions, and technical assistance [3].

IIoT technologies are some of the pillars of the fourth industrial revolution, and they span several fields, including manufacturing, energy, mining, and transportation. In this regard, IIoT can be defined as the combination of smart computing and network technologies applied to industrial processes. This combination seeks the automation of such processes and functionalities within the industry [4]. In this sense, IIoT is indeed improving the productivity, efficiency, safety, and intelligence of industrial factories [5]. Furthermore, it ensures efficient and sustainable production, as a key solution to comprehend the industrial and manufacturing processes.

Services based on IoT, in general, and IIoT, in particular, require the deployment of a large number of devices. On the one hand, these devices need to ensure information and communication reliability. On the other hand, the cost of devices needs to be reduced to afford massive deployments, while access to a power supply cannot be always guaranteed but, in contrast, IIoT devices are frequently battery-driven. Altogether, the need to interconnect a large number of devices with limited hardware capabilities and power resources poses new challenges in terms of energy efficiency, reliability, and security/privacy [6].

In one potential architecture, communication and computation tasks are mostly carried out by devices at the edge of the network, which have constrained resources, including limited battery lifetime. Energy management thus becomes a crucial problem in IIoT [7]. In order to tackle it, some low-power application protocols are used for communication and data transfer in IIoT. Among them, Message Queueing Telemetry Transport (MQTT) has been receiving increased attention from the research and development communities, due to its ease of use, lightweight design, Quality of Service (QoS) capabilities, and easy integration [8,9,10]. MQTT is built on top of the widely used Transmission Control Protocol (TCP), which provides end-to-end reliability, flow control, and congestion management. However, TCP, initially designed for wired networks, exhibits performance issues when used over fast capacity-varying channels such as wireless links [11].

Recently, the Quick UDP Internet Connections (QUIC) protocol has been developed to overcome some of TCP’s limitations and inefficiencies. This protocol is implemented on top of User Datagram Protocol (UDP) and, among other functionalities, it provides flow and congestion control, reliable communications, in-order delivery to upper layers, and multi-stream management. QUIC was introduced by Google [12] and standardized by the Internet Engineering Task Force (IETF) [13]. QUIC is being adopted by many companies due to its good performance: reduced latency, lightweight connection establishment, stream multiplexing, and capability to overcome head-of-line blocking [14].

All in all, the interplay of IoT protocols, such as MQTT, and QUIC will potentially improve the communication performance of IIoT scenarios. In this sense, this paper introduces a methodology to analyze the performance of MQTT when used over QUIC and TCP over a real platform, using commercial off-the-shelf devices. For this, we design and develop a testbed over which we carry out an evaluation of relevant key performance indicators: delay and energy consumption. This assessment does not only shed light on the suitability of QUIC as a transport protocol for IoT traffic, but it also serves to assess the validity of the proposed methodology.

The aforementioned goal is tackled by means of the following phases, which lead to the main contributions of this paper:Design and implementation of a real testbed based on Raspberry Pi devices. The code developed in the framework is publicly available at https://github.com/tlmat-unican/MQTT-QUIC-for-RaspberryPi (accessed on 1 April 2022). The implementation includes:–Integration of MQTT over QUIC and TCP in Raspberry Pi devices;–Synchronized setup using Network Time Protocol (NTP) for reliable measurements;–Automation of procedures (sending, logging, etc.) for trace generation and result gathering;–Emulation of different channel conditions and technologies, using the *traffic control* Linux utility; it embraces varying end-to-end delay, bandwidth, and different loss probabilities.Execution of a measurement campaign over such a platform to understand the performance of MQTT over QUIC, and to assess its suitability for IoT services, comparing it with the traditional TCP/Transport Layer Security (TLS) configuration.

The rest of the paper is organized as follows. Section 2 describes the related work, highlighting how this research stands out from existing papers. Then, Section 3 discusses the main features of MQTT, QUIC, and TCP protocols. Afterwards, in Section 4, we describe the testbed that we designed and implemented, and we depict the configurations that were used to run the experiments, whose results are discussed in Section 5. The paper concludes in Section 6, where we provide an outlook of our future work.

## 2. Related Work

Since it was initially proposed by Google, the behavior of QUIC has been analyzed under different conditions, but only a few works have focused on studying its performance for IoT services. On the other hand, there do not exist many evaluations of latency and energy consumption when IoT devices use QUIC.

A first group of works focuses on establishing the main features of IIoT. In this sense, Mumtaz et al. [15] analyzed the progress, standardization efforts, and challenges of connectivity for the IIoT realm. They identified some of the main connectivity techniques and requirements, and the potential of IIoT to foster the next industrial revolution. Ferrari et al. [16] included the IIoT paradigm in the Industry 4.0 concept, to manage the information generated by sensing devices, which is then processed, exploiting cloud-based solutions. They described and assessed the estimation of the round-trip latency of data transfer between IIoT devices and the cloud, considering different scenarios. Likewise, Kenitar et al. [17] estimated the latency, over different gateways, when transferring data from the edge to the cloud. In this case, the authors adopt MQTT as a data delivery solution. The scope of these works is broader, while our paper particularly focuses on comparing the performance of different transport protocols that are used to support IoT data delivery with MQTT.

A second group of studies compare the behavior of different application layer protocols, to foster their adoption in IIoT scenarios. Mishra and Kertesz conduct in [18] a survey on how MQTT could be exploited for IoT scenarios. Interestingly, their paper does not consider the use of alternative transport protocols, such as QUIC, for MQTT traffic. Akasiadis et al. [19] introduce a platform that leverages the interoperability of various application layer protocols (including MQTT and Constrained Application Protocol (CoAP)). The authors use a real testbed to assess the feasibility of their proposed solution, over which they evaluate the end-to-end delay. However, they do not consider different underlying connectivity conditions, nor alternative transport layer solutions. Pohl et al. [20] analyzed the performance of different IoT protocols, including Advanced Message Queuing Protocol (AMQP), MQTT, and Extensible Messaging and Presence Protocol (XMPP), over a three-tier testbed, by measuring key indicators, such as latency, throughput, bandwidth, and reliability. The results obtained in the study show that, in the considered scenarios, MQTT outperforms AMQP and XMPP in all categories. In the same way, Silva et al. provided in [21] a comparison and evaluation of IoT communication protocols, such as MQTT, CoAP [22], and Open Platform Communications Unified Architecture (OPC-UA) [23]. Seoane et al. compare in [24] the performance of CoAP and MQTT. They also use a real testbed, as well as emulation techniques to consider different channel conditions. In particular, they use the NetEm application to modify the loss rate. However, they focus on security aspects, and they only use traditional transport protocols. Another interesting paper is the one by Viel et al., who propose in [25] an interface to integrate IoT devices with smart grids, based on the CoAP application protocol.

Among the works devoted to IoT, a number of them focused on the MQTT operation, and in particular on the broker node. For instance, Oliveira et al. [26] analyze the performance of different MQTT broker implementations. They use a real platform featuring Raspberry Pi, but they do not compare the performance of different transport protocols. Similarly, Gammes et al. [27] study the behavior of one of the most widely used MQTT broker implementations (Mosquitto) under normal conditions and considering DoS attacks. Although the paper also includes transport layer attacks (i.e., TCP SYN flooding), it does not compare the performance of MQTT over different transport protocols. From a more general perspective, Gheorghe-Pop et al. describe in [28] a benchmark carried out for various MQTT broker solutions. They do not focus on the same performance indicators as the ones we consider in this paper, such as energy consumption or the interplay with transport layer protocols. Mishra et al. also carry out a study in [29], where they assess the performance of different MQTT brokers, under stress circumstances. They conclude that the Mosquitto implementation outperforms the other alternative solutions for most of the parameters. Moreover, Koziolek et al. compare in [30] the performance of three distributed MQTT broker implementations, but the approach was more focused on their usability, CPU performance, reliability, etc.

Other works also focus on the MQTT behavior besides the broker node. For instance, Ebleme et al. [31] assessed the behavior of MQTT in terms of delay, throughput, and energy consumption using Arduino-based nodes as IoT devices. In addition, Katsikeas et al. [32] concluded that MQTT, as a lightweight protocol, is suitable for industrial scenarios. In their evaluation, they paid special attention to data security, introducing a full assessment of potential security issues, and they studied networking features, using a wind park as a real IIoT scenario. Kodali and Valdas proposed in [33] a solution for fire detection, which used MQTT as a communication protocol in a network comprising NodeMCU temperature sensors and Raspberry Pi devices. Michaelides et al. focus on security aspects of MQTT in [34]. They do not assess the delay, but they also use Raspberry Pi in their testbed, as well as an approach similar to ours to characterize energy consumption.

As can be seen, all these works study the performance of application layer protocols for IoT and IIoT. In particular, MQTT stands out as the most widely adopted and analyzed solution. Nevertheless, these IoT studies do not compare the performance of the application layer protocols over different transport solutions, as we do in this work.

On the other hand, we can also highlight some works that have analyzed the performance of the QUIC transport protocol. For instance, the authors of [35] discussed the main features of various solutions, including Stream Control Transmission Protocol (SCTP), Datagram Congestion Control Protocol (DCCP), TCP, and QUIC. Nepomuceno et al. [36] compared QUIC and TCP in terms of the download time, for web traffic, using various websites. Jung and An proposed in [37] an improvement for video streaming and web data services exploiting QUIC, which yielded an increase in the Quality of Experience (QoE). The proposed solution brings a delay reduction, based on an estimation of the average congestion window whenever a new connection is established.

The authors of [38] evaluated the performance of QUIC in terms of packet loss, delay, and jitter. Similarly, Yu et al. [39] studied the behavior of QUIC, focusing on the packet pacing and the congestion control mechanisms. The results evince the benefits of the multi-stream multiplexing QUIC feature.

In addition, different methodologies have been exploited to study the performance of QUIC, including simulation and emulation environments. De Biasio et al. described in [40] a QUIC implementation in the ns-3 [41] simulator, which included its most important features. Furthermore, Kakhki et al. studied the performance of QUIC over an emulated environment [42]. The main distinctive aspect between these works and the one discussed in this paper is that none of them consider the use of QUIC as a transport solution for IoT-based services.

Only a few works have tackled the evaluation of QUIC as a replacement for TCP in IoT or IIoT scenarios. In [43], Herrero analyzed QUIC as an alternative to both TCP and UDP for IoT services based on CoAP. This work first develops analytical models of CoAP performance over UDP, TCP, and QUIC. Then, an experimental analysis is conducted using the Visual Protocol Stack emulator (VPS+). Differently to this paper, we design and implement a testbed that uses real devices, which is afterwards exploited to conduct the evaluation, which considers MQTT, and not CoAP.

Liri et al. [44] explored and analyzed the performance of QUIC over IoT scenarios, comparing it with widespread solutions, such as CoAP, MQTT, and MQTT-SN. The analysis is carried out using a combination of real devices and an emulated environment. In particular, the authors used virtual machines and WiFi connections, to assess the validity of the obtained results. In all cases, delay and losses were emulated. This work has some similarities with ours, since the platform that we introduce herein also emulates the underlying channel conditions. However, Liri et al. do not exploit QUIC as a transport alternative over which application-level protocols are instantiated. Opposed to this, QUIC sockets are directly used to send IoT traffic, so that QUIC is analyzed as a replacement for CoAP, MQTT, and MQTT-SN (which would use TCP underneath), rather than as a replacement of TCP. Hence, the authors did not assess whether the QUIC operation is suitable for the communication paradigms brought by upper protocols in IoT services (i.e., publish–subscribe). In addition, their evaluation does not include energy consumption.

Two of the closest works to ours are those by Fernández et al. [45,46], where the authors assessed the performance of MQTT over QUIC. However, their methodology was different from ours, and they did not foster an evaluation over a real platform. In this sense, all MQTT entities (publisher, subscriber, and broker) do not run over real and independent devices, but they are virtualized and executed in Linux containers. In addition, different wireless technologies and channel characteristics are emulated with the ns-3 network simulator, which is connected to the Linux containers by means of virtual devices (virtual interfaces, virtual bridges, and TAP devices), so that the whole protocol stack of the host is not used. The methodology introduced in this work is based on a platform that uses real devices, while underlying channels are emulated at the host itself, instead of exploiting a simulation framework.

Table 1 summarizes the overlap of existing works with ours, in terms of the main contributions of our paper. In the first column, we account for the works that look at IoT application layer protocols, such as MQTT or CoAP, including those that evaluate their performance under different underlying connectivity situations. Then, in the second column, we identify the works that study QUIC as a transport protocol alternative, either analyzing its performance or comparing it with other solutions, such as TCP. The third column highlights works that use hardware devices for the performance analysis, and the last one identifies works that focus on energy consumption. As can be observed, papers that cover IoT do not usually pay much attention to the transport layer solution, completely leaving aside QUIC. On the other hand, works studying the performance of QUIC focus on traditional Internet scenarios, and not on IoT services. In any case, the table clearly identifies the gaps that this work fills.

All in all, the work presented herein complements and broadens the available state-of-the-art, by targeting the evaluation of QUIC as a transport alternative to support IoT protocols, using a real testbed.

## 3. IIoT Protocols

In this section, we describe the main features of the protocols involved in the evaluation that we discuss later. The aim is to provide a general understanding of their operation, to better tackle the analysis of the performance results in Section 5.

As was already mentioned, MQTT has been widely adopted in the IoT field, in general, and in IIoT, in particular. This protocol usually relies on TCP, which might show shortcomings when used over wireless environments, as well as on TLS, to enable secured end-to-end communications. On the other hand, QUIC, recently standardized by the IETF [13], may be used as an alternative to the aforementioned scheme, to improve the performance of IIoT services. The following sections describe the three involved protocols, paying special attention to the differences between TCP and QUIC.

### 3.1. Message Queuing Telemetry Transport—MQTT

MQTT is a popular application-level protocol based on the publish–subscribe paradigm [47], which usually runs on top of TCP. It has gained relevant popularity for IoT, due to its ease of implementation, small code footprint, bandwidth efficiency, and client decoupling.

The protocol defines three entities: publisher, subscriber, and broker. Connections are established between publisher and broker, and between subscriber and broker. In all cases, brokers play the server role in traditional client/server architectures, while both publishers and subscribers take the client functionality in their connections.

Publishers generate information, whose type is specified by the so-called topic, and send it to the broker. On the other hand, subscribers register at the broker topics they are interested in. Upon receiving messages from publishers, the broker filters them, according to their topic, and forwards them to the interested subscribers. Since MQTT is an application protocol, two independent transport layer connections are established between each pair of entities.

Another key element that makes MQTT an appropriate application protocol for IIoT services is the support of different QoS levels. There are three supported QoS modes [48]:QoS 0 (at most once): the message is sent only once, with no retries. Since there is not any acknowledgment at the MQTT level, there is no delivery guarantee *per se*. Nevertheless, reliable communication can be enforced by lower layers, such as TCP.QoS 1 (at least once): in this case, the message can be retransmitted until the sender receives an acknowledgment. Upon receiving duplicate messages, a flag is set in the corresponding acknowledgment.QoS 2 (exactly once): the message is delivered exactly once. This is ensured with a four-way handshake to ensure that both the original message and its acknowledgment have been correctly received by receiver and sender, respectively. This QoS mode is the slowest one, and it needs, at least, 2 Round Trip Time (RTTs).

As can be seen, the adopted QoS mode might have a direct impact on the energy consumption, and thus the battery lifetime, of IoT devices [49]. In addition, it provides adaptability to unreliable environments, according to the particular requirements of the IoT service [50].

### 3.2. TCP and QUIC

The Transmission Control Protocol (TCP) is the most widespread transport protocol, supporting most of the current Internet services, including those used in IoT environments. It is a connection-oriented solution, offering a reliable service, ensuring in-order delivery, and providing congestion and flow control. In spite of its generalized use, TCP exhibits notable limitations that may affect relevant QoS parameters for IoT. In particular, the congestion control mechanisms usually employed by TCP do not show appropriate performance over wireless links, and they may cause transmission window depletion, thus reducing the transmission rate. Furthermore, TCP single-stream operation could cause HOL blocking, which may lead to longer delays.

Moreover, TCP was not originally designed to appropriately handle different traffic types—for instance, IoT and IIoT services—usually characterized by a bursty nature, leading to short connections [51]. TCP requires the establishment of an end-to-end connection for all different bursts, thus increasing the application delay, due to the inherent latency of the establishment procedure (three-way handshake). The connection delay can be mitigated using extensions such as TCP Fast Open (TFO), which was proposed to avoid the delay induced by the three-way handshake during reconnections, quite frequent in web services. However, the initial connection still requires the complete three-way handshake, and an additional RTT would be needed to establish the TLS connection.

Due to the aforementioned TCP limitations (among others), many updates or alternatives have emerged, with QUIC [13] being one of the more relevant. The main objectives of its design are to improve communication security and reduce transport layer-induced delays, especially in the connection establishment. It has already been thoroughly tested over the years (Google deployed QUIC on many of its servers to support YouTube clients [12]) and up to 34 drafts were published until the recent release of the first IETF specification [13]. Figure 1 depicts the QUIC stack, comparing it with the traditional TCP one, when HTTP/2 traffic is considered at the application layer.

QUIC is built on top of UDP, and it includes the TLS 1.3 protocol to guarantee a secure connection. The embedding of TLS within QUIC allows application data to be sent at the first RTT, provided that the endpoints had been previously connected (0-RTT). Opposed to this, and due to the TCP and regular TLS handshake, the start of data transfer in traditional TCP connections can only take place after 2 RTTs, as was mentioned earlier.

QUIC information is organized in streams, and it integrates multiplexing techniques to avoid the delay caused by Head-of-Line blocking. This way, when a packet is lost in one stream, only the traffic belonging to this stream is affected. On the other hand, packets in other streams are not hindered, because the orderly reception of streams is not required [39], thus reducing the overall delay in lossy environments.

In addition, QUIC brings additional latency reduction, due to its loss detection mechanisms, which include “Early Retransmits” and tail loss probes [52]. These mechanisms use acknowledgment-based detection with a probe timeout to guarantee that acknowledgments are successfully received. There are clear differences between the operation of loss detection solutions used by TCP and QUIC. Among them, we can highlight that QUIC uses packet sequence numbers to avoid the eventual ambiguity that might occur with TCP, where transmission and delivery order might not necessarily coincide.

Furthermore, QUIC features a better upgrading and migration strategy, due to its user-level implementation. Unlike TCP, QUIC is transparent to middleboxes, because it is encapsulated inside the UDP payload. On the other hand, the deployment of TCP enhancements strongly depends on the possibility of updating these middleboxes. While upgrading QUIC might not cause too much trouble, not all devices would be able to simultaneously switch to a newer version. To enable the coexistence of different versions, QUIC includes a version negotiation mechanism, which allows endpoints to decide on the version that they will use for the connection, before it is actually established. This feature allows customization of the protocol with additional features [13].

Figure 2 depicts the protocol stack that is used during our experiments. We use MQTT over the two transport protocols (TLS/TCP and QUIC) and, compared to Figure 1, we do not include the *multi-streaming* feature, since we do not evaluate its behavior in the work described herewith.

## 4. Evaluation Testbed

As has been already mentioned, MQTT is one of the most relevant application protocols for IoT and IIoT services. As was stated in Section 1, the main goal of this paper is the design and implementation of a methodology that exploits a real platform to study the performance of MQTT over QUIC and TCP, in IIoT environments. In particular, we focus on two of the most relevant performance parameters in IIoT services: delay and energy consumption.

Figure 3 shows the testbed architecture, which mimics a layered IIoT system. At the bottom layer, there is a set of IIoT devices (for instance, sensors) that generate data, and send them to a broker, thus taking the MQTT publisher role. Then, the broker forwards the received information to MQTT subscribers, which are used by cloud services to gather IIoT data. All devices are implemented using Raspberry Pi (model 3B), equipped with a Broadcom BCM2837B0 quad-core A53 (ARMv8) 64-bit system, 1.4 GHz, 1 GB LPDDR2 SDRAM, RJ-45, Ethernet 10/100 and WiFi interfaces. As for the operating system, we use the default choice in Raspberry Pi (i.e., Raspbian), so that we can exploit Linux utilities.

The IIoT device embeds an MQTT publisher, and it also logs information about traffic generation and energy consumption. The former is obtained using a high-precision multimeter (Keysight 64465A), able to record the amperage required by the device. The Raspberry Pi devices that implement the publisher and broker roles are connected with a 100 Mbps Ethernet link. Furthermore, Linux utilities are used over such interfaces to emulate different channel conditions, which mimic the ones that would have been observed over wireless links. In particular, we use Traffic Control (TC) https://tldp.org/HOWTO/html_single/Traffic-Control-HOWTO/ (accessed on 1 April 2022) utilities on the Ethernet interface of the IIoT device. This Linux command allows us to change the interface capacity (bandwidth), to add additional delays, and to establish packet error rates.

Although real wireless links (for instance, WiFi) could be easily used, TC utilities over Ethernet connections leverage a tighter control over channel features, allowing us to conduct systematic and repetitive experiments, over the very same conditions. This approach to emulating underlying technologies does not precisely mimic low-level details of the considered wireless technologies, but our interest is to compare the performance of two transport protocols, under the same connectivity conditions, and so the use of TC to change capacity, delay, and packet losses is sensible. Other works have exploited similar procedures, such as [44], where the authors also use TC, or [24], where NetEm allows the authors to emulate different channel conditions. In [45,46,53], the ns-3 simulator is used, but the channel is modeled with a point-to-point link, whose configuration parameters are the ones that we have considered in this paper: capacity, delay, and error rate. It is worth mentioning the Pantheon framework [54], which adopts the Mahimahi [55] link emulator. It also mimics the characteristics of the underlying connectivity, to analyze the performance of different congestion control solutions. In addition, the approach we have followed allows the emulation of wireless technologies that could not be easily used, such as satellite links.

Then, the Raspberry Pi devices that implement the broker and subscriber roles are also connected by means of an Ethernet cable. In this case, we do not add any particular characteristic to the link between broker and subscribers in the cloud, since it mimics a wired connection, with a lower impact on the communication. Nevertheless, it is worth noting that additional features could be easily added to this link as well, such as delays over the transport network.

As can be seen in Figure 3, all entities use a dual TCP/QUIC implementation of MQTT. For this, we use a QUIC implementation in the golang programming language (quic-go version 0.15.1
https://github.com/lucas-clemente/quic-go (accessed on 1 April 2022)).

In addition, we use, for the publisher and subscriber, an MQTT client based on Eclipse-Paho, which can be configured to run over both TCP and the aforementioned QUIC implementation https://github.com/pgOrtiz90/paho.mqtt.golang (accessed on 1 April 2022). As for the broker, we use the Volant MQ https://github.com/VolantMQ/volantmq (accessed on 1 April 2022) implementation (version v0.4.0-rc6). The regular version provides an MQTT (version 3.1) broker over TCP, while QUIC support has been added in a separate branch https://github.com/fatimafp95/volantmq_2 (accessed on 1 April 2022).

We guarantee the synchronization of devices using NTP, taking the broker internal clock as a reference. It is worth noting that synchronization is performed out-band, and it does not thus affect data traffic. Both the NTP server and client are integrated within the broker and publisher/subscriber devices, respectively. Thanks to this synchronization, we are able to precisely establish the transmission delay between receiver and sender, in order to obtain meaningful statistics, such as the average delay or its coefficient of variation, when using TCP and QUIC.

The experimentation has been automated by means of Python scripts. In each experiment, these scripts set the TC configuration to mimic different wireless technologies: WiFi, cellular, and satellite links. In addition, both the QoS used by MQTT, as well as the number of transmitted packets and its sending rate, are also established.

As a result, two log files are generated at the publisher and subscriber, comprising time instants of sending and reception events of each packet, as well as the energy consumption in the Raspberry Pi that mimics the IIoT device (publisher). In addition, on the publisher side, pcap files are also generated to better study the actual number of packets that are injected into the interface. In this sense, it is worth mentioning that the MQTT QoS setup, the particular features of TCP and QUIC (including the congestion control algorithm), and the interplay between application and transport protocols may strongly increase the number of packets that are actually sent in an experiment.

## 5. Performance Evaluation

This section discusses the performance characterization that was carried out over the testbed previously described. We analyze the behavior of MQTT, over both TCP and QUIC, in terms of traffic delay and energy consumption. In all cases, TCP is used together with TLS, so that all the analyzed configurations provide the same functionalities and can be thus fairly compared.

As mentioned before, we use the TC utility to mimic the characteristics of different wireless technologies. Table 2 enumerates the link delay and capacity of the three considered technologies, as well as the different packet loss rates that were used. As can be seen, we consider different channel qualities, from error-free situations to loss rates of 5%, which would reflect a sensible bad channel situation. Previous QUIC simulation-based analysis (not covering MQTT) used similar loss rates—for instance, [45,56,57]. In addition, Table 3 shows the particular configurations that were used for the various experiments whose results are discussed below. In all cases, application data are randomly generated, since we are not interested in sending any particular information, but only in characterizing the time required, for each packet, to reach the destination.

### 5.1. MQTT Traffic Delay

We first measure the delay suffered by the MQTT traffic. We configure MQTT to use the QoS 0 level, to keep the communication as simple as possible. For each configuration, we send 1000 packets with a 1 s interval between two consecutive packets. This low rate ensures that we do not congest the channel, and any delay variation is thus caused by the transport protocol behavior. Although analyzing the delay in higher load conditions might also be interesting, other mechanisms may come into play, making the analysis more complex. In particular, under such conditions, the congestion control mechanisms used in TCP and QUIC would modulate the response of the transport protocol. This evaluation is left for our future work.

First, in Figure 4 we depict the average one-way traffic delay when using TCP and QUIC over the three considered wireless technologies, and for different error probabilities. It is worth noting that the link delay of each wireless technology is different. In order to fairly compare the results, the minimum value of the *y*-axis in each plot corresponds to the link delay of the corresponding technology: 25, 100, and 600 ms for WiFi, cellular, and satellite links, respectively. In general, we can observe that the delay experienced with QUIC is lower than with TCP. In addition, the results evince that QUIC exhibits a rather stable behavior as we increase the loss probability. On the other hand, the delay experienced with TCP increases more abruptly as the loss probability grows.

In this sense, the delay when using TCP over WiFi grows up to >40 ms with a loss probability of 5%, almost doubling the link delay of this technology (25 ms). In the case of the cellular technology, we again observe an additional delay of around 20 ms, reaching ≈120 ms for the highest loss probability. Eventually, for the technology with the largest link delay (satellite with 600 ms), TCP yields an additional delay of above 60 ms when the channel suffers the highest loss probability.

In the case of QUIC, the results are rather different. When using WiFi, QUIC hardly adds 12 ms of delay for the worst channel conditions, and a similar increase is observed for the cellular technology. As for the satellite link, the delay increase observed for QUIC is ≈35 ms, compared to the 60+ ms obtained with TCP.

Then, in Figure 5, we use a boxplot representation to show the statistical distribution of the delay. The figure only shows the results obtained for the WiFi channel, although similar behaviors could be observed with the cellular and satellite setups. Each boxplot represents the 25th and 75th percentiles (upper and lower box limits), as well as the 5th and 95th percentiles with the lower and upper whiskers, respectively. The corresponding median values (50th percentiles) are shown with a bullet. In order to also depict the delay dispersion and understand the average values that were shown before, we include outlier values, which are outside the aforementioned percentiles. It is worth noting that the ordinate axis is split, to better illustrate the range of outlier values, while still appropriately showing the boxplot range.

As can be observed in Figure 5, most delay samples are within a rather stable range, regardless of the transport protocol (TCP or QUIC) and the loss probability. In this sense, the variation in the average delay observed in Figure 4 is explained by the outliers. For both TCP and QUIC, the variability grows with the loss probability, as was also seen for the average value (cf. Figure 4). Moreover, we can observe that the values reached by the delay outliers using TCP are more than twice those observed for QUIC, as was also seen in Figure 4.

The results in Figure 4 evince that QUIC yields a lower delay than TCP for higher loss rates, and it does not jeopardize the performance when the conditions of the underlying connectivity are better. There is, however, a particular case (ideal cellular link) where the results show a delay for QUIC that is slightly higher than the one seen for TCP. In order to thoroughly analyze this particular case, Figure 6 shows the corresponding delay distributions for the two transport protocols. As can be seen, there were a number of experiments whose delay was greater than the 95th percentile (more than the outliers observed for TCP). These outliers cause an increase in the average value, becoming higher than the one for TCP. In any case, this is a very punctual behavior, which was not observed for any other configuration.

Figure 5 evinces the relevant difference in the delay variability when using TCP and QUIC. In order to provide a quantitative metric of such a parameter (*jitter*), Figure 7 shows the delay Relative Standard Deviation (RSD) observed when using TCP and QUIC over the different wireless links. The RSD is defined as the ratio between the standard deviation and the mean value, and so provides a fair comparison of random variables, regardless of their average value. As expected, Figure 7 shows, for all the wireless technologies and transport protocols, that the variability increases with the loss probability. However, the increase depends on the particular setup. In the case of the WiFi channel, the jitter observed for TCP is much higher than the one observed when using QUIC, even over ideal channels. In addition, we can see that the relationship between loss probability and delay variability is more clear for QUIC, whose RSD steadily grows as we increase the packet loss rate. On the other hand, the delay RSD seen when TCP is used does not have a clear relationship with the loss probability, although it is always higher than the one exhibited by QUIC.

Similar results are seen over the cellular link; see Figure 7b. Again, the variability that is observed for QUIC is always below the one seen when TCP is used, and it exhibits a growing trend with the loss probability. On the other hand, the jitter experienced for TCP does not show a clear relationship with the packet loss rate. Differently, the jitter observed for both TCP and QUIC over satellite links shows a clear relationship with the packet loss probability. Indeed, Figure 7c shows that the RSD linearly grows with the error rate for TCP. In the case of QUIC, we can also observe a growing trend, but there is a saturation effect for loss probabilities greater than 3%, after which the variability stops growing.

All in all, the results evince that not only the delay exhibited by TCP is larger than that observed for QUIC, but the variability of such delay is also remarkably lower when using QUIC. In addition, the results show that QUIC’s jitter is more affected by the error rate.

We now analyze the impact that the MQTT QoS level has over the delay. In this case, we only show the results obtained with QUIC, since the differences with TCP-based communications are similar to the ones discussed so far. Figure 8 illustrates the average delay exhibited by QUIC when using the different QoS levels and when varying the error rate. In general, the results evince that the delay is quite similar for the different QoS levels, especially when the error rate is low. For the highest loss probabilities, we observe different trends, depending on the wireless technology. In the case of WiFi, whose initial delay is the lowest one, 25 ms, a higher QoS level yields a lower delay as we increase the error rate, being especially remarkable when the loss rate equals 5%. As for the cellular technology, the trend is similar, although the delay reduction as we increase the QoS level is less relevant. On the other hand, the results obtained over the satellite link, which has the highest initial delay (600 ms), show a different trend. As can be observed, the higher the QoS level, the longer the observed delay.

We complete the delay analysis with Figure 9, where we plot the RSD of the delay observed when QUIC was used as a transport protocol, for the different QoS levels. First, we can observe that, regardless of the adopted QoS level, the delay RSD is rather low, hardly reaching 0.6 in the worst case, compared with the RSD observed when TCP was used, which was above 1.5 in some setups. If we look at the different technologies, we can see that the trend is similar to the average delay behavior (see Figure 8). In the case of WiFi (lowest link delay, 25 ms), increasing the QoS level has a positive impact on the RSD, particularly for higher error rates. This trend is less evident for the cellular technology, whose initial delay is 100 ms. Finally, a rather different behavior was obtained over the satellite link, with an initial delay of 600 ms. Indeed, Figure 9c evinces that high QoS levels have a negative impact over the *jitter*, for the satellite link, when the error rate grows.

Altogether, the analysis shows that the use of higher MQTT QoS levels may have a positive impact when low-latency technologies are used. On the other hand, for technologies with longer RTTs, such as satellite, higher QoS levels actually lead to worse performance as the error rate grows.

### 5.2. Energy Consumption

As was mentioned before, one of the most critical performance indicators to be considered in any evaluation within the context of IIoT is energy consumption. In order to characterize it, we discuss now the results that were obtained for MQTT over QUIC and TCP. We exploited a high-precision digital multimeter (Keysight 64465A) in our experimental campaign. As in the previous section, we always sent 1000 packets per experiment.

We first study the impact of the packet length and Inter-Arrival Time (IaT), which is defined as the time elapsed between two consecutive packets at the sender. Using the multimeter, we measure the total energy consumed, and we divide it by the time required to send all packets, to obtain comparable measurements of the average power, in Joules per second (J/s). Figure 10 depicts the average power obtained with both TCP and QUIC for 3 different packet lengths (50, 100, and 500 bytes) and IaT values of 1, 10, and 20 ms. In order to see the direct effect of TCP and QUIC over the energy consumption, the results shown in Figure 10 were obtained over an ideal channel, with no losses, delay, or capacity limit. It is worth recalling that TCP was used together with TLS, so that the impact that encryption processes may have on the energy consumption (as well as on the delay) would be similar for all cases. First, Figure 10 evinces that the packet length has not a remarkable impact on the energy consumption. Similarly, the power values obtained when using the different IaT configurations are rather stable, being slightly larger for the lowest one (1 ms). This effect could be a consequence of the sampling process used by the multimeter, although a deeper analysis of this effect will be tackled in our future work. If we compare the results when using TCP and QUIC, we can observe that both protocols yield similar performance in terms of energy consumption. In fact, the results show almost identical power levels for packet lengths of 50 and 500 bytes. On the other hand, for 100 byte packets, TCP yields a slight power reduction.

Finally, we have broadened the energy consumption analysis by studying the number of actual transmissions when using MQTT over both TCP and QUIC, and for the different wireless technologies that were previously analyzed. In this case, we use an IaT of 1 s, to avoid triggering congestion control mechanisms, as we did in the previous characterization. However, due to the memory limitations of the multimeter, we could not monitor the consumed energy during the whole experiment’s duration. Instead, we use the generated pcap files to count the overall number of packets sent at the physical layer. Although this approach does not allow us to know the actual energy consumed in each measurement, it enables a fair comparison between TCP and QUIC in terms of energy consumption, since it would be proportional to the number of transmissions.

Figure 11 shows the number of packets transmitted per second when using TCP and QUIC for the different wireless technologies. The results evince again a clear impact of the underlying wireless link over the observed performance. In the case of the WiFi channel (lowest link latency), TCP requires a higher number of transmission events as we increase the error rate. On the other hand, the behavior of QUIC is more stable, and it stays around 4 packets per second, regardless of the error rate. Similar behavior is observed over the cellular link, where the number of transmissions for QUIC is again quite stable, while TCP causes more transmissions at the physical layer when the packet loss rate is higher. Surprisingly, the results observed over the satellite link, in Figure 11c, show that the number of transmission events that are required for both protocols decreases as the error rate becomes higher. This effect is a consequence of the slow increase in the congestion window, due to the long RTT. This behavior, and the impact of different congestion control mechanisms over the energy consumption and the overall system performance, will be addressed in our future work.

The results show that there might be a small penalization when using QUIC in terms of energy consumption for some of the configurations that we have evaluated (satellite). However, the same conclusion might not be valid for different scenarios—for instance, considering other application protocols (i.e., CoAP). In this sense, it would be necessary to develop more generic energy models—for instance, exploiting complexity-related approaches.

## 6. Conclusions and Future Work

In this paper, we have analyzed the performance of one of the most widespread application protocols for IIoT, MQTT, and its interplay with different transport protocols: TCP and QUIC. In this sense, we have seen that, despite its growing relevance, there do not exist many works that have analyzed the performance of QUIC as a transport protocol alternative for IoT traffic. This work sheds light on whether it is a suitable alternative for MQTT-based applications.

Furthermore, we introduce an evaluation methodology, which entails the design and development of an evaluation testbed, comprising Raspberry Pi devices, able to provide delay- and energy-related metrics. Over such a framework, we have carried out an extensive measurement campaign to analyze the performance of MQTT/QUIC and MQTT/TLS/TCP stacks over a variety of channel conditions, which emulate different wireless technologies.

The obtained results show that the adoption of QUIC as a transport solution may bring relevant benefits in terms of delay over all the analyzed wireless technologies. In this sense, we observed a delay reduction of 25.5% over WiFi links when the loss rate was 5%. For cellular and satellite connections, characterized by longer RTTs, the reduction is less relevant, but QUIC still yields lower delays (4.5 and 5.1%, respectively).

More interestingly, the use of QUIC also leads to a sharp jitter (delay variation) mitigation, especially over error-prone links. We studied the RSD of the delay, and the reduction that was observed in our experiments was 61%, 42%, and 41%, for the WiFi, cellular, and satellite links, respectively, when the loss rate was 5%.

Focusing on the interplay between MQTT and QUIC, we have observed that the use of MQTT QoS levels has a positive impact in terms of delay over some particular scenarios, especially those having shorter RTTs. On the other hand, the energy measurements obtained with a high-precision multimeter evince that QUIC does not significantly increase the energy consumption. We have actually seen that both TCP and QUIC generate a comparable amount of traffic.

Based on the results of the measurement campaign carried out over a platform featuring commercial off-the-shelf devices, we can thus conclude that QUIC is indeed a suitable transport protocol alternative for typical IoT scenarios.

In our future research, we will broaden the work presented here in different ways. First, we will extend the testbed to include other application protocols, such as ZeroMQ, CoAP, or OMA LwM2M. In addition, we will add more complex channel emulators, such as *NetEm* or *MahiMahi*, to mimic different channel conditions. We will also analyze the impact of using other congestion control solutions, such as BBR, and their suitability for IIoT scenarios. Finally, we will exploit the proposed methodology and the implemented platform to develop more generic energy models, which could be applied to a wider range of configurations.

## Figures and Tables

**Figure 1 sensors-22-03694-f001:**
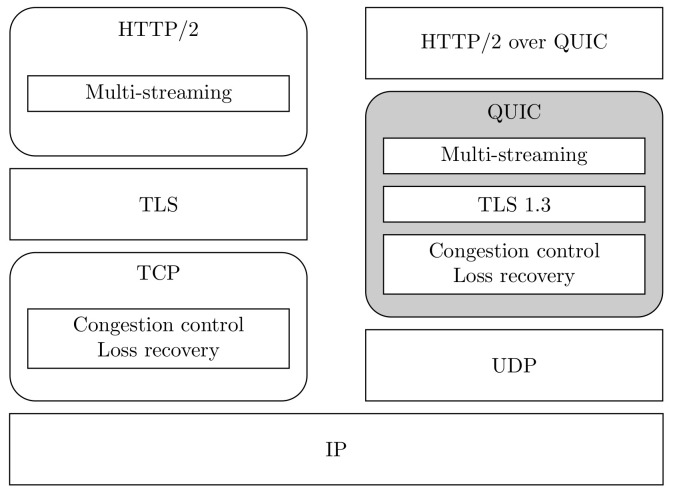
HTTP/2 over TCP and QUIC.

**Figure 2 sensors-22-03694-f002:**
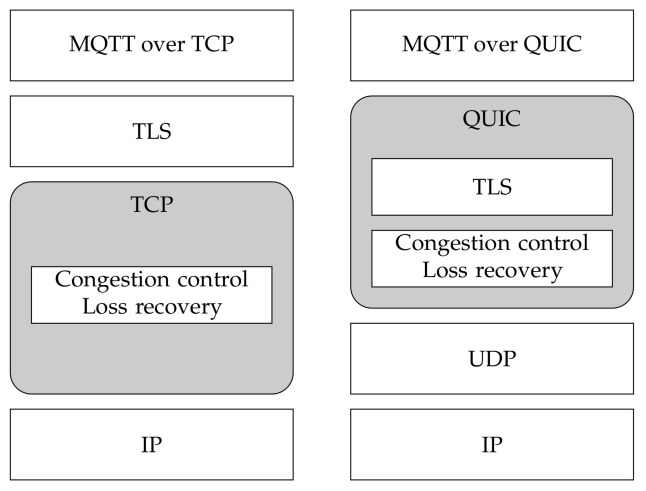
MQTT over TCP and QUIC.

**Figure 3 sensors-22-03694-f003:**
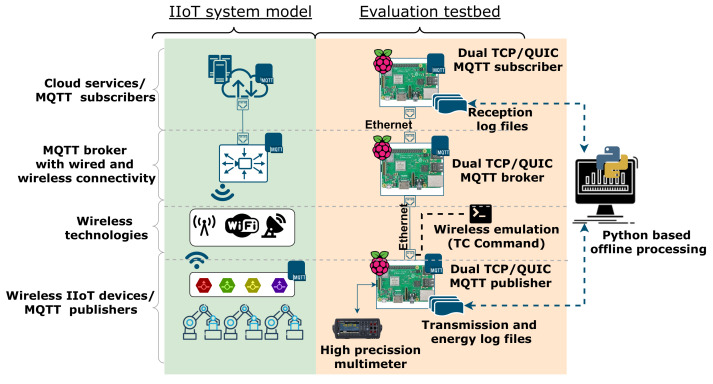
Setup diagram: Raspberry Pi devices taking MQTT roles.

**Figure 4 sensors-22-03694-f004:**
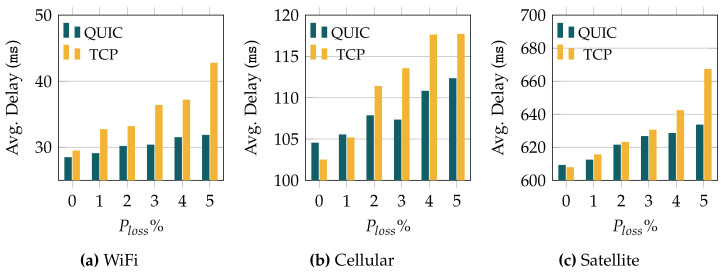
Average delay observed for QUIC and TCP, and different loss probabilities.

**Figure 5 sensors-22-03694-f005:**
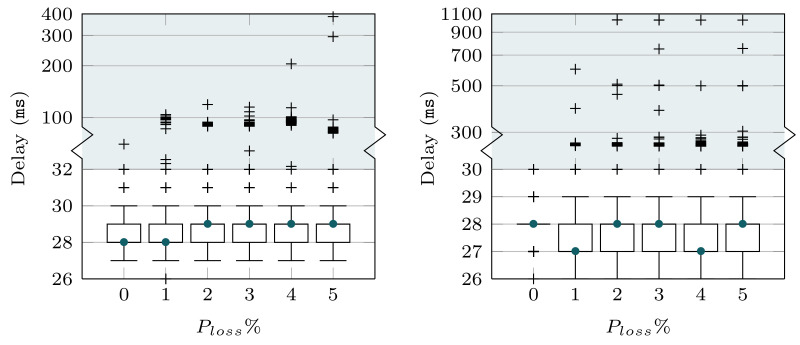
Boxplot of the delay experienced with QUIC and TCP over the WiFi channel. Ordinate axis is split to show the range of outliers, which are shown using a logarithmic scale. (**a**) QUIC (**b**) TCP.

**Figure 6 sensors-22-03694-f006:**
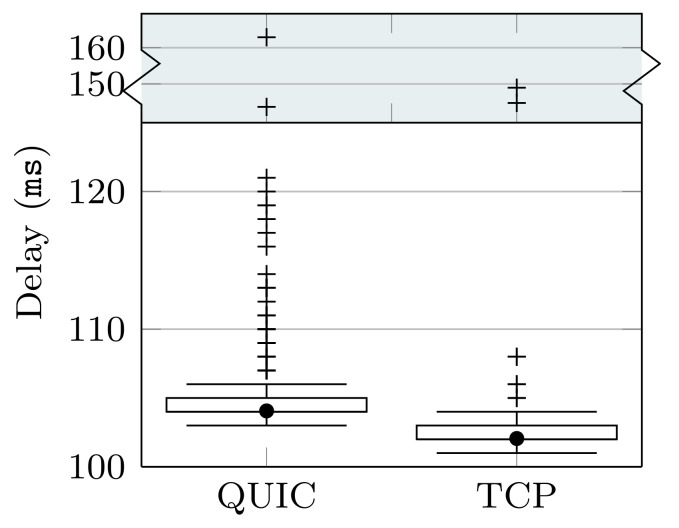
Boxplot of the delay experienced with QUIC and TCP over the cellular channel with ideal conditions ($P_{loss}=0$). Ordinate axis is split to show the range of outliers, which are shown using a logarithmic scale).

**Figure 7 sensors-22-03694-f007:**
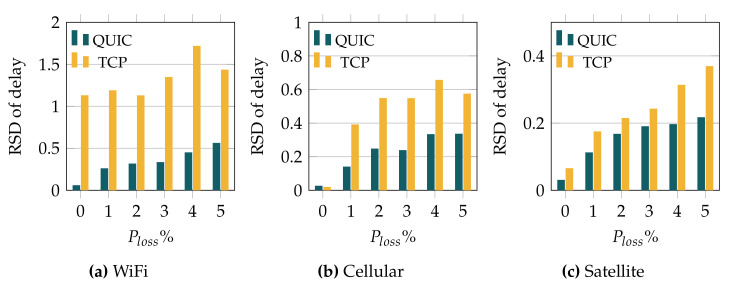
Delay RSD observed with QUIC and TCP over the different wireless technologies.

**Figure 8 sensors-22-03694-f008:**
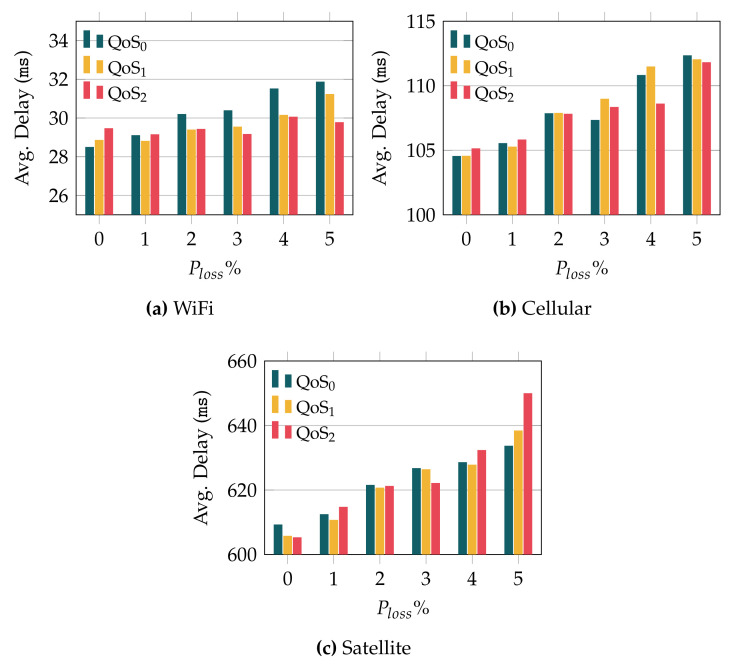
Delay experienced with QUIC for the various QoS configurations.

**Figure 9 sensors-22-03694-f009:**
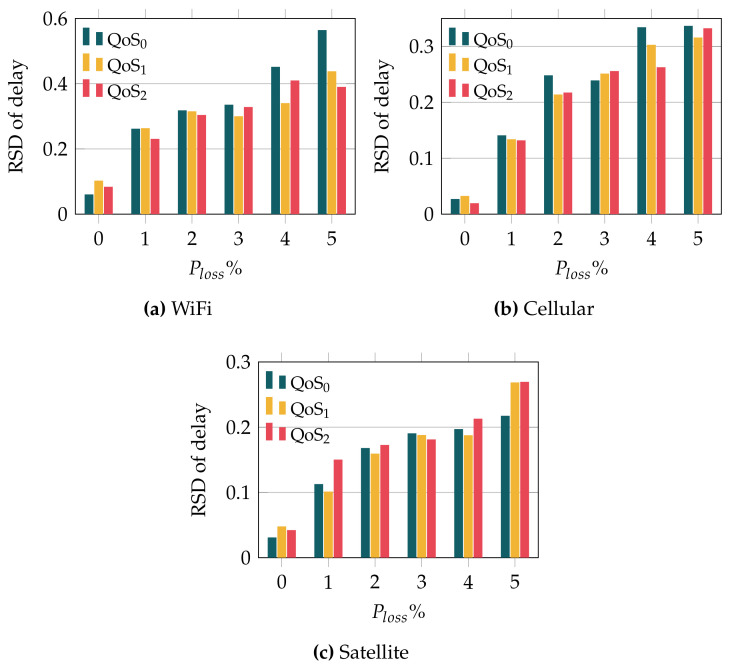
RSD of MQTT traffic over QUIC and for the different MQTT QoS levels.

**Figure 10 sensors-22-03694-f010:**
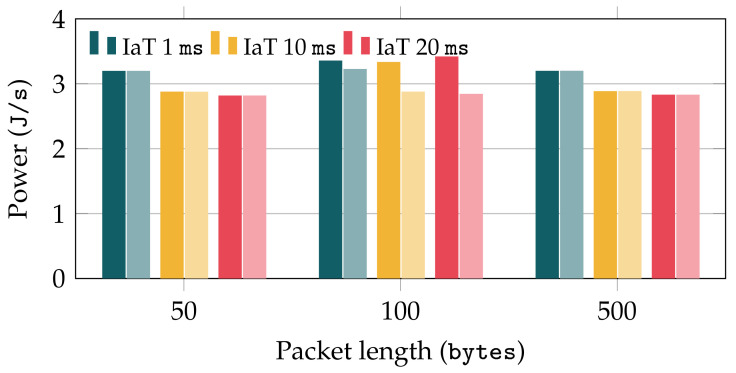
Power consumption of QUIC (left dark bars) and TCP (right pale bars) for different packet lengths and inter-arrival times.

**Figure 11 sensors-22-03694-f011:**
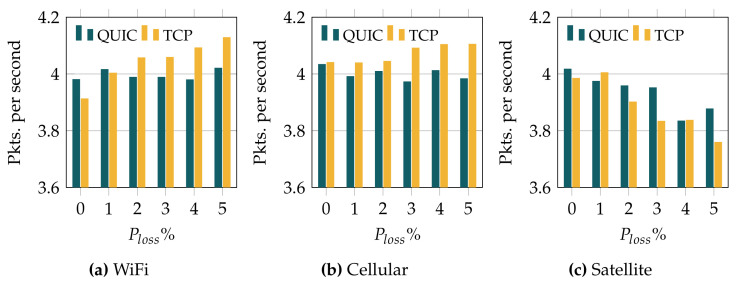
Sending packet per second of MQTT over QUIC and TCP.

**Table 1 sensors-22-03694-t001:** Features covered by the related works.

	IoT Application Protocols	QUIC as Transport Solution	Over Hardware Devices	Energy Consumption
[18]	✓	✗	✗	✗
[19]	✓	✗	✓	✗
[20]	✓	✗	✓	✗
[21]	✓	✗	✓	✗
[24]	✓	✗	✓	✗
[26]	✓	✗	✓	✗
[31]	✓	✗	✓	✓
[34]	✓	✗	✓	✓
[35]	✗	✓	✗	✗
[36]	✗	✓	✗	✗
[37]	✗	✓	✗	✗
[38]	✗	✓	✗	✗
[39]	✗	✓	✗	✗
[42]	✗	✓	✗	✗
[43]	✓	✓	✗	✗
[44]	✗	✓	✓	✗
[45,46]	✓	✓	✗	✗

**Table 2 sensors-22-03694-t002:** Network parameters for different technologies.

	WiFi	Cellular	Satellite
Delay (ms)	25	100	600
Capacity (Mbps)	20	10	1.5
Loss rate (%)	[0,5]	[0,5]	[0,5]

**Table 3 sensors-22-03694-t003:** Testbed configurations.

Figures	Packet Length (B)	Packet Interval (ms)	# of Packets	MQTT QoS
MQTT traffic delay				
Figure 4, Figure 5, Figure 6 and Figure 7	100	1000	1000	0
Figure 8 and Figure 9	100	1000	1000	0, 1, 2
Energy consumption				
Figure 10	50, 100, 500	1, 10, 20	1000	0
Figure 11	100	1000	1000	0

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
