# Peer review of "Delay and Energy Consumption of MQTT over QUIC: An Empirical Characterization Using Commercial-Off-The-Shelf Devices"

_sensors, 2022, doi:10.3390/s22103694_

Round 1

Reviewer 1 Report

In this paper, it discussed the QUIC protocol and did some tests on the system performances in terms of delay ,jitter and energy consumption. 
These results indicated that QUIC with MQTT could improve the system performance. It is a good work, suggest to accept it with minor revision.

Check the spelling and redraw some Figures so that they have a much better presentation.

Author Response

Dear referee,
in the attached file we include our responses to your comments. In addition, apart from the updated version of the manuscript, we include in this submission another document that highlights the changes performed in this review round.

Reviewer 2 Report

the article "Delay and energy consumption of MQTT over QUIC: An empirical characterization using commercial-off-the-shelf devices" has an interesting theme but several points are unclear

In the introduction, what is the novelty of the work developed?
I suggest highlighting this, for example by comparing or commenting on a recent work 2022. In fact, there is only one work published this year, out of a total of 48.

The theme that involves the work developed is constantly being updated and, therefore, it is necessary to keep it as updated as possible.

In section 2. Related work
It is not clear the research problems discussed and which the authors intend to solve in relation to the researched literature.

There are no specific criteria and parameters defined in this section.

I strongly suggest a comparative table listing the items mentioned above.

Do they present news in relation to what is already published, for example, other protocols?
This work was published in the same journal, I believe it can help the authors in several gaps in the work developed: https://doi.org/10.3390/s20102849

Compare research problems in relation to literature, consider articles published in the last 3 years (2020, 2021, and 2022);

In section 4. Evaluation testbed
Pack sizes, the exact amount of tests with different sizes and quantities are not clear. The presented graphics can be more noticeable if the authors include a code snippet, so the reader can repeat it in other scenarios.

I also suggest deepening the conclusions.
It is not clear the scientific contributions obtained from the tests and results.

References
Many works do not have DOI or ISSN, more information needs to be included.

Author Response

(The authors gave the same response as above.)

Reviewer 3 Report

The paper presents the results of an empirical study of delay and energy of MQTT over QUIC protocol for Industrial IoT. The topic is of interest for the readers of Sensors and the paper provides a valuable contribution.

The overall quality of the paper is good and I recommend its publications with the following minor improvements.

  1. I would suggest to use references as such, without using them as part of the text. For instance, at line 95, instead of "... the work presented in [6] analyzed..." I would say "... Xu at al. analyzed ... [6]."
  2. In addition to the representation of the HTTP/2 over TCP and QUIC protocol stack (Figure 1), I would provide a graphical representation of MQTT over QUIC.
  3. I recommend to add a reference to the paper published on Sensors by Fernandez et al. in 2021.
  4. A careful read is required to find typos that are not automatically found (e.g., row 460: "specially" should be "especially"; row 367: "interesting" should be "interested")
  5. The boxplot of Figure 5 should refer to the same conditions of the first bars of the bar graph in Figure 3.b, but the average values are not the same.

Author Response

(The authors gave the same response as above.)

Round 2

Reviewer 2 Report

The authors made improvements, for the final version I suggest reviewing the journal's format, not including the Short Biography of Authors.
Revise text and better value your work by describing the results obtained, etc. in the conclusions.

Author Response

Dear referee,

we again include in the attached file our responses to your comments. Besides, as in the previous review round, we include in this submission a supplementary document highlighting the changes we have performed.

This manuscript is a resubmission of an earlier submission. The following is a list of the peer review reports and author responses from that submission.

Round 1

Reviewer 1 Report

Minor comments:
  - I will suggest not to use and define abbreviatures in the Asbtract.
  - Once you define abbreviatures, please use them (i.e. Industrial IoT).
  - Define only once each abbreviature.
  - Fix grammar errors like "in order delivery", "on the the delay", "acknowledge-ment have been correctly received by the received and sender", "In the our testbed"... There several of them.
  - I will suggest you to review some expressions like "As can be observed, all the aforementioned works adopt MQTT, and assess its performance in different scenarios, and from different angles."
  - Lines 171-178 could be removed because they are not relevant.
  - Paragraph in lines 179-185 is confuse. A better description must be done.
  - Try to avoid repeated, twice or more times sentences or ideas like "In addition, the single-stream nature of TCP would then lead to the HOL effect, which might induce much longer delay."
  - Remove ")" in the URL "https://github.com/VolantMQ/volantmq)"

Major comments:
  - The contributions you mention in the paper must be refined because, as an example, the measure of performance is not a research contribution. Moreover you must mention which is new in the emulation of a particular implementation of QUIC you proposed.
  - You wrote "Although this work shares some of our objectives,...", but you never explained your objecvies previously in the paper. You only state in Section 4 "In this paper, our main goal is to assess the perfor-mance of MQTT when used over QUIC and TCP for IIoT services." (I will suggest you to write that in the Introduction Section to state better the objectives of your paper). 
  - In general you did not review properly, but only presented superficially, others works in the state of the Art Section. I.e. you wrote "but it is directly used to send the IoT traffic." That sentence must be clearly explained becuse it is understandable. Plese review properly other works.
  - It seems that reference [36] is a work previously and recently worked by the same authors of this proposal paper. Please make an effort to differentate both works and explain clearly if you only present here an emulation (of a particular implementation of QUIC on a particular kind of Raspberry Pi running unknown communication interfaces) of previous simulated work. 
  - What you wrote in lines 216-219 is not strictily correct, because TCP could accept different implementations and configuration that do not force 3-way handshaking. Please reconsider that sentenc and contemplate that in your implementation for comparing with QUIC.
  - Section 3 is a comparison between TCP and QUIC in a very wide range, but the paper will be more appropriated if you only center the comparison of them from the energy saving and the latency saving that MQTT could provide being implemented over them. Please focus that Section and provide analytical results that you could verify in your practical implementation.
  - Which is the reason to consider only a 5% of packet loss in every technology (Figure 3), is it equally probable to loss packets in the different technologies?
  - How do you explain the behaviour of QUIC when Delay is greater than the TCP one in Cellular and Sattelite?
  - Which is the length of each packet for measuring delay?

Reject comments:
  - It seems that reference [36] is a work previously and recently worked by the some same authors of this proposal paper. Please make an effort to differentate both works and explain clearly if you only present here an  extension of that work emulating previous novel proposal on RaspBerry Pi.
  - It is very important you to declared clearly what kind of communication interface used between Raspberry Pies in Figure 2 in order to clearly the reader could reproduce the experiment.
  - Related to the above comment, Figure 2 specifies clearly that you emulated three different kind of wireless link using Linux TC tool (I suppose that it is available in RaspBerry Pi 3B also, but you must explain what operating system used in the Raspberry Pies, and if you had problems...). But what you did was to vary the speed of communication in the phisycal communication interface among Raspberry Pies. So it is not clear the study you did could explain the things you wrote about WiFi, Cellular and Sattelite communications that implies very complex processes that you did not take into account and represent the important facts of performance of QUIC and TCP on those networks. Please do a more complex emulation and discuss properly the results including a discuss Section.
  - Results are based in a particular packet rate and sizes (for energy consumption tests) and lmited amount of Raspberry Pies. But what about different lengths of packets and packet rates. I suggest you to model formally the problem (considering important parameter of communications) and then to emulate it considering complex model for WiFi, Cellular and Sattelite communications to obtain realistic and representative results and verifying your model.

Author Response

Dear reviewer, in the attached document we give detailed answers to the your comments, and we also introduce the changes that were made in the manuscript, which are highlighted in the revised document.

Reviewer 2 Report

This paper presents performance comparisons between MQTT/QUIC and MQTT/TCP with a real testbed. Metrics include traffic delay and energy consumption. Raspberry Pi is used as nodes. Network condition (WiFi, Cellular, etc.) is done by emulation. Eclipse-Paho and Volant MQ as MQTT implementation and quic-go as QUIC implementation.

Strength:

A real testbed is used for testing

Energy consumption is considered as well

Weakness:

Only one kind of device (Raspberry Pi) is used.

Network condition is simulated.

Questions & Suggestions:

  1. In several experiments, it mentioned “send 1000 packets with a 1 second interval between two consecutive packets”, however, more explanations are needed. For example, what’s the length of the packets? Do they include the extra packets that including the handshake packets in TCP? Does the QUIC header count in the packet length?
  2. Figure 4 is not very clear. For example, where is the lower quartile, and median?
  3. In Section 5.2, the measured energy consumption is due to only packet transmission, or due to CPU, memory as well? Since QUIC includes decryption, so it should consume more energy for CPU.

Author Response

Dear revieer, in the attached document we give detailed answers to your comments, and we also introduce the changes that were made in the manuscript, which are highlighted in the revised document.

Reviewer 3 Report

This paper presents an experimental test of the QUIC protocol. Implementing protocols on commercial devices is interesting. In this word to main issues in IoT networks are explored. 

Energy consumption should be evaluated as computational complexity. Measure the current in each device is a good approximation but, in the opinion of this reviewer, this information can be wrong if readers implement their ideas in different testbeds. The O notation presents a general approximation of the complexity of the protocols. The results of O notation will be the same in several electronic devices. 

QUIC should be defined in the abstract in a similar way as IoT or MQTT.

Figure 2. Cloud services/ MQTT subscribers  by MQTT subscribers.

Table 1--Loss(%) --- double dot in 0..5

Author Response

Dear reviewer, in the attached document we give detailed answers to your comments, and we also introduce the changes that were made in the manuscript, which are highlighted in the revised document.

Round 2

Reviewer 1 Report

Minor comments:
  - I will suggest not to use and define abbreviatures in the Asbtract. DO NOT USE ABBREVIATURES IN THE ABSTRACT.
  - Define only once each abbreviature.
  - Try to avoid repeated, twice or more times sentences or ideas like "In addition, the single-stream nature of TCP would then lead to the HOL effect, which might induce much longer delay." SEEMS to be more.

Major comments:
  - The contributions you mention in the paper must be refined because, as an example, the measure of performance is not a research contribution. Moreover you must mention which is new in the emulation of a particular implementation of QUIC you proposed. Your answer does not change my considerations (see other comments about that).
  - You wrote "Although this work shares some of our objectives,...", but you never explained your objecvies previously in the paper. You only state in Section 4 "In this paper, our main goal is to assess the perfor-mance of MQTT when used over QUIC and TCP for IIoT services." (I will suggest you to write that in the Introduction Section to state better the objectives of your paper). THE ACTION YOU REFER MUST BE WRITTEN EXPLICITLY.
  - It seems that reference [36] is a work previously and recently worked by the same authors of this proposal paper. Please make an effort to differentate both works and explain clearly if you only present here an emulation (of a particular implementation of QUIC on a particular kind of Raspberry Pi running unknown communication interfaces) of previous simulated work. I THINK IT IS NOT ENOUGTH TO PRESENT HERE A DIFFERENT IMPLEMENTATION OF YOUR TESTBED PLATFORM (NOT ENOUGTH CONTRIBUTION).
  - Section 3 is a comparison between TCP and QUIC in a very wide range, but the paper will be more appropriated if you only center the comparison of them from the energy saving and the latency saving that MQTT could provide being implemented over them. Please focus that Section and provide analytical results that you could verify in your practical implementation.
  - Which is the reason to consider only a 5% of packet loss in every technology (Figure 3), is it equally probable to loss packets in the different technologies? PLEASE JUSTIFY IN THE PAPER.
  - How do you explain the behaviour of QUIC when Delay is greater than the TCP one in Cellular and Sattelite? ACTIONS ARE REQUIERED TO WELL STABLISH THAT PERFORMANCE (PLEASE INCLUDE IN THE PAPER YOUR EXPLANATION CAREFULLY).
  - Which is the length of each packet for measuring delay? PLEASE GVE MORE DETAILS ABOUT THE COMPOSITIVION OF THOSE 100 B.

Reject comments:
  - It seems that reference [36] is a work previously and recently worked by the some same authors of this proposal paper. Please make an effort to differentate both works and explain clearly if you only present here an  extension of that work emulating previous novel proposal on RaspBerry Pi. EXPLANATION DOES NOT SOLVE MY QUESTION PROPERLY. PLEASE ANSWER PROPERLY.
  - It is very important you to declared clearly what kind of communication interface used between Raspberry Pies in Figure 2 in order to clearly the reader could reproduce the experiment.
  - Related to the above comment, Figure 2 specifies clearly that you emulated three different kind of wireless link using Linux TC tool (I suppose that it is available in RaspBerry Pi 3B also, but you must explain what operating system used in the Raspberry Pies, and if you had problems...). But what you did was to vary the speed of communication in the phisycal communication interface among Raspberry Pies. So it is not clear the study you did could explain the things you wrote about WiFi, Cellular and Sattelite communications that implies very complex processes that you did not take into account and represent the important facts of performance of QUIC and TCP on those networks. Please do a more complex emulation and discuss properly the results including a discuss Section. YOU ANSWER: "Since the goal is to compare the behavior of the two transport protocols, this methodology is valid." WHICH IS NOT CORRECT AS YOU RECOGNIZE "Although
it is true that the specific mechanisms implemented by lower layers (MAC and PHY) have an impact over such performance, we have simplified the features of such technologies, and we just vary the capacity as well as the RTT.", RECOGNIZING MY COMMENTS. IT IS IMPORTANT TO STATE CLEARLY WHAT YOU DID (AN EVALUATION OF QUIC OVER RASPBERRY PI EHTERNET COMMUNICATIONS VARYING SPEED OF COMMUNICATION: FROM THAT TO SAY YOU EVALUATED QUIC OVER THE WIRELESS COMMUNICATIONS, THERE IS A LONG UN-RESEARCHED TRIP. PLEASE TAKE NOTE OF THAT AND MODIFY THE PAPER AS SUGGESTED.
  - Results are based in a particular packet rate and sizes (for energy consumption tests) and lmited amount of Raspberry Pies. But what about different lengths of packets and packet rates. I suggest you to model formally the problem (considering important parameter of communications) and then to emulate it considering complex model for WiFi, Cellular and Sattelite communications to obtain realistic and representative results and verifying your model. IF YOU WILL CONSIDER A DETAILED ENERGY SAVING EVALUATION PLEASE MODIFY THE PAPER IN ORDER NOT TO MENTION THAT YOU HAVE TESTED ENERGY CONSUMPTION. AND PLEASE RECONSIDER THE CLEAR PRESENTATION OF WHAT IS NEW IN THIS PAPER IN RESPECT TO YOUR PREVIOUS WORK.

Author Response

We provide answers to the referee's comment in the attached file.

Reviewer 2 Report

My questions are answered well.

Author Response

Thank you for your help to improve the manuscript.